# Health-related stigma among Indigenous Peoples in Canada: A scoping review

**Liam Rourke** ⓘ *ᵒ, **Ronald Damant**ᵒ, **Janice Y. Kung** ⓘᵒ, **Chantell Widney**ᵒ

Faculty of Medicine and Dentistry, University of Alberta, Edmonton, Alberta, Canada

ᵒ These authors contributed equally to this work.
* lrourke@ualberta.ca

## Abstract

### Background

Indigenous communities in Canada are disproportionately affected by health conditions linked to stigma, warranting the attention of researchers seeking to understand this culturally-determined phenomenon. This study explores the scope of research on health-related stigma conducted with the First Nations, Inuit, and Métis Peoples.

### Method

We conducted a scoping review using the method described by Arksey and O'Malley. We searched health and social science databases from 1963 to present using the subject headings *Stigma* and *Health* delimited by terms indexing over 600 Indigenous groups in Canada. Within the 1,852 results, we searched for reports in which the construct *stigma* was used to describe some facet of the participants' experience of a health condition. We excluded studies in which stigma derived from the participants' sexual orientation, occupation, or cultural identity. We extracted information about the participants' health condition, Indigenous affiliation, forms of stigma experienced, and their responses.

### Results

25 studies involving 1,187 participants met our inclusion criteria. Inuit, First Nation, and Métis participants were drawn from communities in Alberta, British Columbia, Manitoba, Nunavut, Ontario, Quebec, and Saskatchewan. Stigma was reported by people living with HIV, mental health concerns, tuberculosis, STIs, type 2 diabetes, arthritis, physical disabilities, asthma, arthritis, substance use disorders, and FASD. Most frequently they reported enacted stigma expressed as social and physical distancing by perpetrators who interpret the diagnoses and symptoms as marks of social deviance or disease contagion. The primary response to stigma was to conceal one's condition in ways that increased the disease burden.

### Interpretation

Canada's Indigenous communities have escaped the attention, or perhaps interest, of researchers investigating health-related stigma. In five decades of research, the subject

**Data availability statement:** All relevant data are within the paper and its Supporting information files.

**Funding:** This study was funded by the Canadian Institute of Health Research (185352 to RD).

**Competing interests:** The authors have declared that no competing interests exist.

surfaces only tangentially in reports designed to explore other aspects of their health. In the absence of research, pressing questions remain, some about stigma as a construct of social science and some about the health of Indigenous Peoples in Canada.

## Introduction

Health researchers have turned to the construct *stigma* for insight into puzzling patient behaviors, including a reluctance to get tested, adhere to treatments, or confide in those who might provide support [1]. Stigma has been described as an attribute that predisposes one to disdain, and researchers have found that the attributes include a growing list of health conditions, including HIV/AIDS [2], mental health concerns [3], substance use disorders [4], lung cancer [5], obesity [6], and, most recently, COVID-19 [7] and post-COVID-19 condition [8].

A fuller explanation of stigma lists its co-occurring components: labeling, stereotyping, separation, status loss, and discrimination [9]. To work an example, the diagnosis *HIV-positive* can evoke derogatory generalizations of those living with the condition such as *promiscuous* or *reckless* which are dehumanizing and engender discriminatory attitudes and behavior [10]. Avoiding any of the components of stigma motivates the patient behaviors that concern health professionals: Those with COPD are hesitant to use their oxygen in public [11]; those with mental health concerns are reluctant to confide in family or friends [12], and those with COVID-19 symptoms are apprehensive about getting tested [13]. These types of responses to the aversive effects of stigma add to the burden of illness, increase morbidity and mortality, and imperil public health.

Each of stigma's co-occurring components are expressed in ways that reflect the social norms and expectations of particular communities; therefore, researchers explore how health-related stigma varies between culturally distinct groups. Numerous studies have found, for example, that people living in rural communities are more reluctant to disclose their health status than those living in urban settings. Explanations include differences in the frequency of face-to-face interaction, levels of anonymity, and the presence or absence of a moral consensus [14]. Similarly, researchers have shown that people living in societies categorized as *collectivist* experience more intense stigma than people living in *individualist* societies. Here, theorists point to variations in the interdependence of community members, the strength of agreement on norms, and the commitment to sanctioning violators [15–17]. Researchers have also found that health-related stigma varies with beliefs about the etiology of disease. Those attributing illness to supernatural causes are more likely to distance themselves from people with health problems than those attributing illness to biological causes [18,19]. In these ways, socio-cultural factors have an explanatory role in several facets of health-related stigma.

Some of these factors have been used to differentiate Indigenous from non-Indigenous communities in Canada. For instance, StatsCan [20] estimates that 60–80% of the nearly two million First Nations, Métis, and Inuit Peoples in Canada live in rural settings while the estimates for non-indigenous Canadians living in such settings is 18%. Many of the communities are also characterized as collectivist, particularly in regard to healthcare practices [21]. As one Indigenous author explains: "At the heart of our beliefs, the care and wellness of the collective is given more credence than individual needs (p. 16) [22]. Regarding the findings that health-related stigma is more prevalent in communities that see a link between illness and supernatural forces, many of the Indigenous groups in Canada have holistic concepts of the etiology of disease and include spiritual actions among the list of causes [21–25]. The beliefs and practices of the Indigenous communities in Canada are not homogenous–no more so than those of the country's non-indigenous communities; however, there may be patterns in the way that health-related stigma is triggered, expressed, experienced, and mitigated.

A discussion of health-related stigma experienced by Indigenous People living in Canada occurs alongside ongoing discussions of institutional stigma and structural racism experienced by First Nations, Inuit, and Métis Peoples. Depictions of the former appear in the Canadian Medical Association's Apology for Harms to Indigenous People [26], which expresses shame for issues such as the substandard care provided in 'Indian Hospitals', medical experimentation, and coerced sterilization. The latter is detailed in the Truth and Reconciliation Committee's investigation and report of equally egregious practices in the context of the residential school system [27].

These discussions focus on the discrimination of one ethnic and cultural group by a more powerful other. Our investigation, building on Goffman's germinal work, investigates the labels, derogatory generalizations, and malicious behaviors that take place within culturally and ethnically homogenous communities. Our objective is to determine the scope of research on health-related stigma conducted with the Indigenous Peoples living in Canada. Specifically, we seek to determine the extent to which the literature is representative of: i) the First Nations, Inuit, and Métis communities dispersed across Canada's expansive geographic and political regions, ii) the set of health conditions that have been linked to stigma and affect Indigenous communities, iii) the types of stigma experienced by people with various health conditions, and iv) their strategies for managing the stigma they encounter.

## Methods

To accomplish these objectives we conducted a scoping review using the methodology described by Arksey and O'Malley [28] and codified in the PRISMA extension for scoping reviews (PRISMA-ScR) [29]. (See S1 Table).

### Search strategy and selection criteria

We searched MEDLINE, CINAHL, PsychINFO, Sociology Abstracts, and Google Scholar for articles published from 1963–the year Goffman published his germinal work on stigma–to September 4, 2024. We used the subject headings *Stigma*, *Health,* and *Illness* delimited by terms indexing over 600 Indigenous groups living in Canada. (See S2 Table). This returned 1,852 results. Within these we searched for reports in which the term *stigma* or its constituent processes –labeling, stereotyping, separation, and discrimination–were used by the participants to describe, explain, or interpret some facet of their experience of a defined health condition. We excluded studies in which:

- the participants were not Indigenous Peoples living in Canada, or the data collected from this group was not separable from the data gathered from others;

- the participants' experiences of stigma stemmed primarily from qualities other than their health condition; most frequently, these included the participants' occupation, sexual orientation, gender, and cultural or ethnic identity. Stigma that arises at the intersection of these factors is a pressing topic for research; however, it was not the focus of this review.

We included only primary research (both post-positivist and interpretivist), and excluded articles categorized as essays, commentaries, position statements, letters, or reviews.

### Data extraction

From the reports we extracted information about:

- the province, territory, city, or community from which participants were recruited

- the Indigenous group with which the participants' identified

- the health condition that qualified participants for inclusion in the study

- the forms of stigma experienced by the participants (see Table 1)

- the beliefs and motivations driving stigma, and

- the participants' responses to stigma

## Data screening, abstraction, and mapping

The first author screened all titles and abstracts, and a second author screened 25-percent of those results to clarify the decision-making process. Discrepancies were resolved through discussion among the authors. The first-author screened all of the full-texts that passed abstract review, and a second author independently screened 10-percent of the full texts as a final measure of the reliability.

We developed a data extraction form in Google Sheets, which included each article's title, authorship, number of participants, area of recruitment, Indigenous affiliation of the participants, health condition, forms of stigma experienced, drivers of stigma, and responses to stigma.

## Results

Our broad search returned 1,852 unique studies. Of those, 997 underwent fulltext screening (See S3 Table.), and 25, involving 1,187 participants, met our criteria and were included in the review. (See Fig 1). The characteristics of each study are presented in Table 2. Below, we summarize the data regarding five aspects of the studies: 1) the representativeness of Indigenous Peoples in Canada, 2) the health conditions for which stigma was reported, 3) the forms of stigma experienced by people with various conditions, 4) the drivers of stigmatizing attitudes and behavior, and 5) the participants' responses to stigma.

### Representativeness

1,187 Indigenous Peoples living in Canada were interviewed or surveyed in this literature. The largest number were Inuit (n=435) followed by First Nations (n=303) and Métis Peoples (n=40). An additional 409 participants were identified only as Aboriginal (n=140) and Indigenous (n=269). Participants were recruited from seven of the country's thirteen provinces and territories; these were Alberta, British Columbia, Manitoba, Nunavut, Ontario, Saskatchewan, and Quebec.

### Health conditions

The largest number of studies was conducted with participants living with HIV (n=10) and mental health concerns (n=5). The remaining studies recruited participants living with

**Table 1. Forms of health-related stigma.**

| Form | Description |
| --- | --- |
| Enacted | Concrete instances or actions of discrimination that an individual faces due to their health condition. |
| Anticipated | The expectation an individual has regarding negative reactions, discrimination, or disapproval they might face due to their health condition. |
| Self | The internalization of negative stereotypes and societal attitudes towards individuals with health issues. |
| Secondary | Public disapproval or negative perceptions that individuals face due to their association with a stigmatized person. |

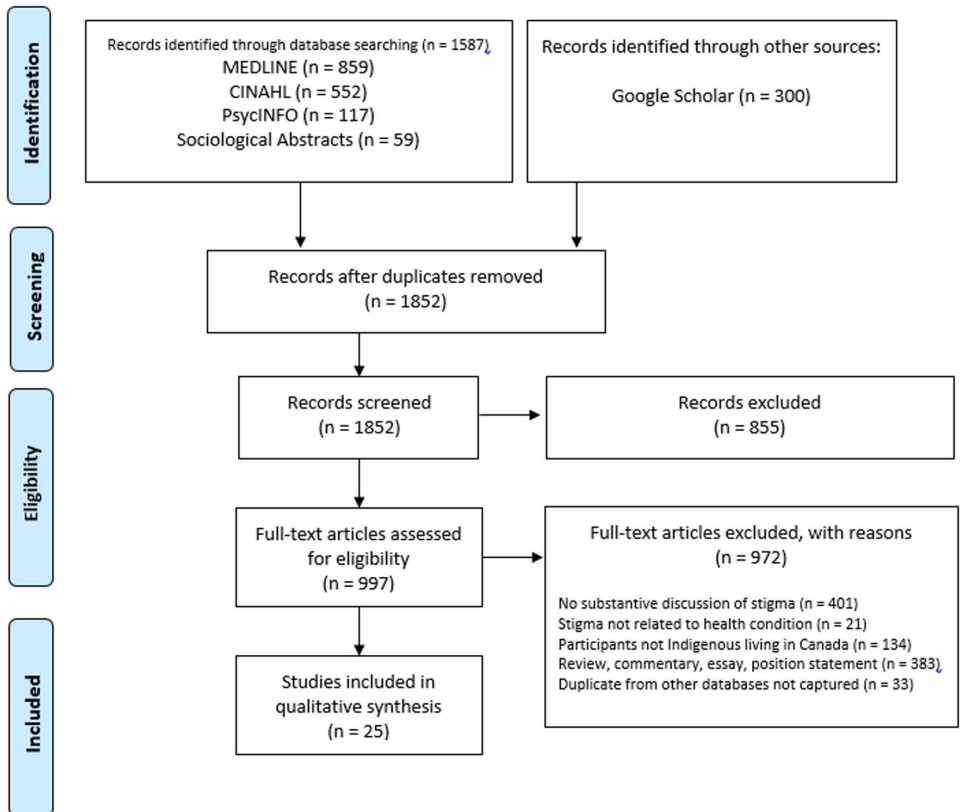

**Fig 1. PRISMA flow diagram.**

tuberculosis (n=3); sexually transmitted infections (n=2), type 2 diabetes (n=2), physical disabilities (n=1), arthritis (n=1), asthma (n=1), substance use disorders (n=1) and fetal alcohol syndrome disorder (n=1).

## Forms of stigma

Participants described four forms of health-related stigma: enacted, anticipated, self, and secondary. Enacted stigma was the most common, with participants in 17 of the 25 studies experiencing this type of discrimination. It took several forms, one of which was social distancing as portrayed by the following participant: "Once you disclose your condition, you lose your family and your friends–even your children can desert you. When I told friends they started avoiding me [31]. Physical distancing was also common, specifically for those with contagious conditions. A participant with tuberculosis described an encounter while boarding a flight: "The pilot told me to sit in the back, and he said, 'Don't touch anything.' He wouldn't handle my bag, and when I tried to give him my documents, he held out a paper bag and said, 'Put them in there'" [45]. Another expression of enacted stigma involved participants being blamed for their health condition. A participant with type-2 diabetes explained: "People are like, 'Stop eating junk food; stop eating so much!' I barely eat junk food. And they say, ' It's your fault you have diabetes because you're a lazy ass'" [54]. Other expressions of enacted stigma included labeling, stereotyping, and trivializing symptoms.

Anticipated stigma was the second most common form of health-related stigma, surfacing in 15 of the 25 reports. The participants' descriptions of this experience parallelled their

**Table 2. Characteristics of included studies.**

| Author Year | Number of participants, Indigenous affiliation[1] | Area of recruitment [2] | Forms of stigma [3] | Drivers of stigma | Response to stigma |
|---|---|---|---|---|---|
| **People living with HIV (PLHIV)** | | | | | |
| Cain 2013 [30] | 55 First Nations<br>11 Métis<br>2 Inuit<br>4 Other | Saskatchewan | Anticipated | Not discussed | Conceal condition |
| Donnelly 2016 [31] | 8 Aboriginal | Vancouver | Enacted<br>Self | Contagion avoidance<br>Norm enforcement | Conceal condition |
| Flicker 2007 [32] | 48 First Nations<br>7 Métis<br>6 Inuit | Quebec<br>Ontario | Enacted<br>Anticipated<br>Self | Contagion avoidance<br>Norm enforcement | Conceal condition<br>Withdraw socially |
| Hatala 2018 [33] | 21 Indigenous | Canada | Enacted<br>Self | Contagion avoidance<br>Norm enforcement | Conceal condition<br>Withdraw socially |
| Hillier 2021 [34] | 29 First Nations | Ontario | Enacted | Contagion avoidance<br>Norm enforcement | Conceal condition |
| Hillier 2023 [35] | 29 First Nations | Ontario | Enacted | Norm enforcement | Conceal condition |
| Loutfy 2012 [36] | 52 Aboriginal | Ontario | Enacted<br>Anticipated<br>Self | Norm enforcement | Conceal condition |
| Mill 2010 [37] | 10 First Nations<br>3 Métis<br>2 Inuit<br>1 Unknown | Edmonton<br>Ottawa | Enacted<br>Anticipated | Contagion avoidance<br>Norm enforcement | Conceal condition |
| Woodgate 2017 [38] | 21 Indigenous | Winnipeg | Enacted | Contagion avoidance | Conceal condition |
| **Mental Health Concerns** | | | | | |
| Ferrazzi 2017 [39] | 165 Inuit | Arviat<br>Iqaluit<br>Qikiqtarjuaq | Enacted<br>Anticipated | Norm enforcement | Avoid treatment |
| Gibbons 2007 [40] | 89 Indigenous | Canada | Enacted<br>Self | Norm enforcement | Avoid treatment<br>Withdraw socially<br>Conceal condition |
| Isaak 2020 [41] | 115 First Nations | Manitoba<br>Saskatchewan | Enacted<br>Anticipated | Norm enforcement | Conceal condition |
| Kirmayer 1997 [42] | 137 Inuit | Povungnituk<br>Salluit | Enacted | Norm enforcement | Conceal condition |
| Litwin 2019 [43] | 49 Inuit | Baker Lake, Cambridge Bay, Grise Fiord, Hall Beach, Igloolik, Kugaaruk, Kugluk-tuk, Naujaat (Repulse Bay), Taloyoak, Qikiqtarjuaq, Reso | Enacted<br>Anticipated | Norm enforcement | Avoid treatment<br>Conceal condition |
| **Tuberculosis** | | | | | |
| Komarnarski 2016 [44] | 20 First Nations | Prairie Provinces | Anticipated | Contagion avoidance | Avoid treatment |
| Mayan 2019 [45] | 112 Indigenous | Alberta<br>Manitoba<br>Saskatchewan | Enacted<br>Anticipated<br>Self | Contagion avoidance | Withdraw socially |
| Moller 2010 [46] | 29 Inuit | Nunavut | Enacted<br>Anticipated<br>Self | Norm enforcement | Conceal condition |
| **Sexually Transmitted Infection** | | | | | |

*(Continued)*

**Table 2.** (Continued)

| Author Year | Number of participants, Indigenous affiliation[1] | Area of recruitment [2] | Forms of stigma [3] | Drivers of stigma | Response to stigma |
|---|---|---|---|---|---|
| Corosky 2016 [47] | 25 Inuit | Arviat | Enacted Anticipated Self | Norm enforcement | Avoid treatment Conceal condition |
| Rusch 2008 [48] | 51 Aboriginal | Vancouver | Enacted Anticipated | Norm enforcement | Not discussed |
| **Fetal Alcohol Spectrum Disorder** | | | | | |
| Aspler 2022 [49] | 19 Indigenous | Canada | Enacted | Norm enforcement | Conceal condition |
| **Substance Use Disorder** | | | | | |
| Borland 2013 [50] | 29 Aboriginal | Ontario | Enacted Self | Norm enforcement | Conceal condition |
| **Wheelchair User** | | | | | |
| Croxoll 2020 [51] | 18 Mohawk | Akwesasne Six Nations | Enacted Anticipated Self | Norm enforcement | Avoid treatment Withdraw socially |
| **Arthritis** | | | | | |
| Loyola-Sanchez 2020 [52] | 52 Blackfoot | Siksika Nation | Enacted Anticipated | Norm enforcement | Avoid testing Conceal condition |
| **Asthma** | | | | | |
| Stewart 2013 [53] | 26 First Nations 19 Métis 1 Inuit | Alberta | Enacted | Norm enforcement | Avoid testing Conceal condition |
| **Type-2 Diabetes** | | | | | |
| Wicklow 2012 [54] | 7 Indigenous | Manitoba | Enacted Self | Norm enforcement | Not discussed |

Notes:

[1.] The authors' categorizations of their participants' indigenous affiliation at the most specific level.

[2.] The most specific level of information available regarding the geographic or political region from which participants were recruited.

[3.] Each form of stigma is described in Table 1.

descriptions of enacted stigma except they pointed to expected behaviors (rather than experienced). As one participant said of their health status: "I want to tell people, but I'm just really afraid of how they're going to react or what they're going to say" [31]. The core of this experience involved the projection of unkind attitudes and beliefs onto others. The anticipation stemmed from discriminatory behavior that had been observed toward others with the same illness. One participant explained, "I wasn't ready to address [my health condition] because I'd seen others being shunned, and it was awful" [32].

The final two forms of stigma were self stigma, in which participants blamed themselves for their illness, and secondary stigma in which family members of those with a health condition experienced stigma. Self stigma was described by participants in nine of the studies while secondary stigma was reported in two.

## Drivers

Most frequently, stigma was attributed to an urge to demarcate or enforce a community's values, and it stemmed from the belief that illnesses befall those who violate their community's moral order. In one instance, a person's pill-rolling tremor was attributed to a history of

stealing, and in another, a tuberculosis outbreak in a family was attributed to the misbehavior of a member of the family [33]. This was cited as the driver of stigma by people living with HIV, type-2 diabetes, mental health concerns, and mobility impairments, and participants with these health conditions said they endured the morally loaded labels 'sinful,' 'blameworthy,' 'gluttonous," undeserving," slothful,' 'promiscuous,' 'burdensome,' and 'dirty' [31,33–35].

Contagion avoidance was identified as a driver of stigma too, and it was associated with the two infectious health conditions that appeared in this literature, HIV and tuberculosis. One participant explained, "When the community finds out someone is HIV-positive they boot them out because they think that they're going to catch it from that person" [53]. A participant living with tuberculosis explained, "When people in my community hear the word 'TB' they're all, 'Oh my God' because they're afraid of TB. They're scared they might catch it. Friends and community members will distance themselves from them over these fears" [45].

### Responses to stigma

When participants experienced stigma, their principal response was to conceal their health condition, and they had three ways to accomplish this: They withheld their health status from family, friends, and acquaintances; they withdrew socially from the larger community; and they avoided presenting publicly for treatment. One participant describes their disclosure concerns: "I didn't tell any of my coworkers. I didn't want them to shoo me away or talk about me behind my back. I keep it to myself because I just don't want people to know" [33]. Others provided insight into their reluctance to engage in testing and treatment: "It is very difficult for people showing signs of depression to see the mental health nurse because they don't want to be labeled as 'crazy'. Enrolling in treatment creates the risk that the community will become aware that a person's illness has been formally recognized and labeled" [35]. Another participant added, "… people take their medication secretly because they don't want others to know they have a disease. People might start to think, 'aah this person has the disease" [33]. In many of the communities from which participants were recruited, the forums of testing and treatment can inadvertently disclose one's health status.

### Discussion

The purpose of this study was to determine the scope of research on health-related stigma experienced by Indigenous Peoples in Canada. We found the scope narrow in several respects, including the range of health conditions studied, the depth of interpretation of the collected data, and the extent to which this group was represented. Only 25 studies have been conducted with this population over a 51-year period in which stigma has been a productive lens for investigating a significant social determinant of health.

The range of health conditions the authors studied was narrow, with only 10 appearing in this set of studies. People living with HIV and people with mental health concerns appeared most frequently, and that is consistent with their prevalence in the larger literature. Yet, other conditions that generate substantial investigation; such as COPD, obesity, lung cancer, epilepsy, COVID-19, and post-COVID-19 condition; did not appear at all. These are among the most prevalent diseases affecting Indigenous communities and regularly linked to health-related stigma.

The depth of interpretation in the studies was superficial. Of the 25 studies that met our inclusion criteria, only six indexed the extensive body of literature on stigma to formulate research questions, design methods, and structure interpretations of interview and survey data. Most of the studies were open-ended explorations of their participants' experiences of a health condition, and in these, the subject of stigma arose extemporaneously. When it did,

authors did not engage with the literature to compare their findings to existing research, substantiate or contradict the conclusions of previous studies of stigma, test the assumptions of health-related stigma models, or, in general, to build on existing research.

Finally, the extent to which the population is represented is uneven. The numerous distinct communities in Canada's five Atlantic provinces have no representation, nor do the Indigenous Nations, tribes, and communities of the Yukon and the Northwest Territories. Conceivably, the unique histories, languages, and cultural practices of these groups influence the way in which a sociocultural phenomenon such as stigma takes shape. Regarding those who were included, participants in ten studies were described only as *Indigenous* or *Aboriginal*. This wrongly implies something homogenous about the approximately 1.8 million Indigenous people in Canada, and it is suggestive of the processes of labeling and stereotyping that are constituent of stigma. At the same time, it obscures most of the variables that are pertinent to insightful investigations of health-related stigma.

Despite its narrow scope, a pattern of health-related stigma is visible in these studies. Beginning with the belief that illness is a mark of social deviance or that symptoms signal contagiousness, perpetrators distance themselves–both socially and physically–from those with a variety of health conditions and work to devalue and discredit them. Seeking to avoid this aversive social behavior, the ill, in turn, conceal their symptoms, diagnoses, and treatments in ways that contribute to, not subtract from, the burden of disease.

In most aspects, this description of stigma is consistent with the model generated through interviews and observations in non-Indigenous communities. Beginning with Goffman's [55] construction of stigma, illness has been framed as a form of deviance, and perpetrators orient toward the ill as they would anyone violating their community's norms. Researchers consistently point to norm enforcement as an impetus for the stigma directed toward those with the health conditions appearing this review, including people living with sexually transmitted and blood borne infections [56], mental health concerns [57], type II diabetes [58], and persons using wheelchairs [59], asthma [60], tuberculosis [61], arthritis [62], and substance use disorders [63].

The second justification for stigma that emerged in the review was contagion avoidance, and this too finds substantial support in the literature. Theorists have argued for an evolutionary advantage in keeping one's distance from those bearing any sign of disease, and, consistent with our findings, it was frequently indexed in the stigma toward people living with HIV [64] and tuberculosis [65].

Canada's Chief Public Health Officer has called for more research on health-related stigma experienced by Indigenous peoples in Canada [66]. A better appeal might specify research that builds on the expansive body of theory, instrumentation, and working hypotheses that have been developed in the previous five decades of research. Subsequent studies might also attend more earnestly to the heterogeneity of Indigenous peoples in Canada. Rather than describing participants flatly as *Aboriginal* or *Indigenous*, authors might describe facets of their sample that are pertinent to pressing questions on health-related stigma. This could include cultural beliefs about illness held by a tribe, band, or community; histories with infectious diseases such as tuberculosis and smallpox; access to healthcare; healing traditions, and any norms, values, and customs that influence the social construction of various illnesses.

This body of literature is too small and exploratory to substantiate suggestions for practice. One issue that surfaced in many reports, however, warrants consideration. Healthcare providers worry about stigma's relationship with the inclination to avoid testing and treatment. Several participants explained that the settings of testing and treatment inadvertently provide cues to one's health status. Because concealment is their principal means of managing stigma, any perception of increased disclosure risk is a significant concern. Further, it is heightened

among this population, many of whom were recruited from small communities with limited healthcare facilities. Efforts to address this issue may reduce the reluctance to be tested and to participate fully in one's treatment, thereby decreasing the burden of a stigmatized health condition.

## Supporting information

**S1 Table. PRISMA-ScR checklist.** This is the preferred reporting items for systematic reviews and meta-analyses extension for scoping reviews checklist.
(DOCX)

**S2 Table. Full search strategy.** This is the full search strategy as executed in MEDLINE.
(DOCX)

**S3 Table. Articles assessed for eligibility.** This is the complete list of 997 articles that underwent fulltext screening.
(XLSX)

## Author contributions

**Conceptualization:** Liam Rourke, Ronald Damant, Janice Y. Kung, Chantell Widney.

**Data curation:** Liam Rourke, Ronald Damant, Janice Y. Kung.

**Formal analysis:** Liam Rourke, Ronald Damant, Janice Y. Kung.

**Funding acquisition:** Liam Rourke, Ronald Damant.

**Investigation:** Liam Rourke, Ronald Damant, Janice Y. Kung, Chantell Widney.

**Methodology:** Liam Rourke, Ronald Damant, Janice Y. Kung, Chantell Widney.

**Project administration:** Liam Rourke, Ronald Damant.

**Resources:** Ronald Damant, Chantell Widney.

**Supervision:** Liam Rourke, Ronald Damant.

**Writing – original draft:** Liam Rourke, Ronald Damant, Janice Y. Kung, Chantell Widney.

**Writing – review & editing:** Liam Rourke, Ronald Damant, Janice Y. Kung, Chantell Widney.

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
