## [Decision Letter · Decision Letter 0]

22 Dec 2024

PONE-D-24-51625Health-Related Stigma Among Indigenous Peoples in Canada: A Scoping ReviewPLOS ONE

Dear Dr. Liam Rourke,

Thank you for submitting your manuscript to PLOS ONE. After careful consideration, we feel that it has merit but does not fully meet PLOS ONE’s publication criteria as it currently stands. Therefore, we invite you to submit a revised version of the manuscript that addresses the points raised during the review process.

We look forward to receiving your revised manuscript.

Kind regards,

Julia Morgan

Academic Editor

PLOS ONE

Journal Requirements:

2. Please ensure that your PRISMA flow diagram is included in your main manuscript file as Figure 1; please see the PLOS ONE submission guidelines for systematic reviews and meta-analyses at https://journals.plos.org/plosone/s/submission-guidelines#loc-systematic-reviews-and-meta-analyses .

3. In this instance it seems there may be acceptable restrictions in place that prevent the public sharing of your minimal data. However, in line with our goal of ensuring long-term data availability to all interested researchers, PLOS’ Data Policy states that authors cannot be the sole named individuals responsible for ensuring data access (http://journals.plos.org/plosone/s/data-availability#loc-acceptable-data-sharing-methods).

4. Thank you for stating the following in your manuscript:

“This study was funded with a grant from the Long COVID Web.”

“Long COVID Web”

5. Please include captions for your Supporting Information files at the end of your manuscript, and update any in-text citations to match accordingly. Please see our Supporting Information guidelines for more information: http://journals.plos.org/plosone/s/supporting-information .

6. As required by our policy on Data Availability, please ensure your manuscript or supplementary information includes the following:

Reviewers' comments:

Reviewer's Responses to Questions

**Comments to the Author**

1. Is the manuscript technically sound, and do the data support the conclusions?

Reviewer #1: Yes

Reviewer #2: Yes

2. Has the statistical analysis been performed appropriately and rigorously?

Reviewer #1: N/A

Reviewer #2: N/A

3. Have the authors made all data underlying the findings in their manuscript fully available?

Reviewer #1: Yes

Reviewer #2: Yes

4. Is the manuscript presented in an intelligible fashion and written in standard English?

Reviewer #1: Yes

Reviewer #2: Yes

5. Review Comments to the Author

Reviewer #1: The article presents a compelling examination of health-related stigma among Indigenous populations in Canada. The introduction is well-structured and effectively articulated; however, it would benefit from further enrichment through additional references that contextualize the socio-cultural dimensions of stigma and its profound impact on the health outcomes of these populations. The study objectives are clearly delineated, and the methodology is robust; however, the inclusion of a PRISMA flow diagram to illustrate the study selection process would enhance the clarity of the methodological approach. Additionally, the article would be strengthened by the inclusion of a conclusion section, which would offer a synthesis of the key findings, along with a discussion of the study's limitations. It would also be valuable to incorporate recommendations for future research in this area to guide subsequent investigations and foster deeper understanding of the subject matter.

Reviewer #2: Thank you for this informative and well written article. I have the following minor suggestions:

1. In the introduction, I would like a bit more rationale for why you chose to focus on Indigenous Peoples in Canada, and not other countries.

2. For international readers, you may want to include in the introduction about the inter-generational trauma experienced by this group and how this could impact both health and stigma and health-seeking behaviours.

3. For the inclusion criteria, you stated you included primary research articles, but did you have any restriction on methods of the primary research articles (i.e. only qualitative articles)? If you did, this should be stated n the inclusion criteria. If you looked at any methodology that was primary research, this should be stated in the inclusion criteria.

4. Regardless of the answer for question 3 above, I would like the type of methods to be specified in the data that were extracted, and information about methods of the included studies to be included in the table of articles.

5. I would like a table title and number for all the tables in the manuscript.

6. I wonder if any of the articles included in the synthesis had a definition of stigma: if some did, this should be included in the data extraction. If none of them had a definition of stigma, this should be stated.

6. PLOS authors have the option to publish the peer review history of their article (what does this mean? ). If published, this will include your full peer review and any attached files.

**Do you want your identity to be public for this peer review?** For information about this choice, including consent withdrawal, please see our Privacy Policy .

Reviewer #1: **Yes: ** Dr. Manisha Gore

Reviewer #2: No

---

## [Author Response · Author response to Decision Letter 1]

14 Jan 2025

Thank you for the close and thoughtful reading of our manuscript and for the offering suggestions that have improved the work and our ability to communicate with readers. Below, we respond to each of the 16 suggestions we received in the following order: Editor (items 1 - 8) > Reviewer 1 (items 9 - 11) > Reviewer 2 (items 12 - 16 with a reference to item 9). Your comments are italicized, our responses follow. Please note that the information provided here duplicates the information in our attached document "response to reviewers". Its format may be easier to read.

Dear Editor

We appreciate the thoughtful suggestions from you and your reviewers. They have improved our manuscript. Below, we detail the revisions.

Please ensure that your manuscript meets PLOS ONE's style requirements.

Completed.

Please ensure that your PRISMA flow diagram is included in your main manuscript file as Figure 1.

We have had considerable difficulty positioning the PRISMA flow diagram in the main manuscript, so we have added it as a separate word document and identified its position in the main manuscript in the hope that the skilled staff at PLOS ONE can assist with this task.

Please also provide non-author contact information (phone/email/hyperlink) for data access.

We have now included all of our data as Supplemental Information. The only data we did not submit originally was a table presenting all of the fulltext articles that were assessed for inclusion, along with the decision to include or exclude, and the data that was extracted from each of the included articles. That table is now available as an excel file named S1 Fulltext articles assessed for inclusion, and listed in the Supplemental Information section of the main manuscript.

Please remove any funding-related text from the manuscript.

We removed the text from the manuscript and identified our funders appropriately within the Editorial Manager submission forms.

Please include captions for your Supporting Information files at the end of your manuscript.

We have presented our titles and captions for our three supporting information files (p. 30), titled them appropriately, added captions, and updated the corresponding in-text citations correctly (p. 6-7).

Please ensure your manuscript or supplementary information includes a numbered table of all studies identified in the literature search, including those that were excluded from the analyses. List the reason(s) for exclusion.

We included a table of all 997 studies assessed for inclusion, the decision that was made for each, by whom, when, and the data that was extracted (from the articles that were included). The table is included as a Supplement. (S3 Table. Articles assessed for eligibility.)

If applicable for your analysis, a table showing the completed risk of bias and quality/certainty assessments.

Assessments of this nature are not required or conventional in scoping reviews. It is one quality that distinguishes scoping reviews from systematic reviews.

Please review your reference list to ensure that it is complete and correct.

We reviewed and updated our reference list so that all in-text citations are included in the references list, the in-text citations are numbered consecutively, and the appropriate square-bracket format is used.

The manuscript would benefit from additional references that contextualize the socio-cultural dimensions of stigma and its profound impact on the health outcomes of [Indigenous Peoples living in Canada].

In the final section of the Introduction (last paragraph of page 5) we added a brief discussion of the systematic racism and institutional stigma experienced by First Nations, Inuit, and Métis Peoples, and index the Canada’s Truth and Reconciliation Commission and the Canadian Medical Association’s 2024 apology for harms to Indigenous Peoples . We note too that our review focuses on the labelling, stereotyping, and discrimination that stems specifically from one’s health condition. This is in contrast to other discourses whose concern is the stigma that arises from gender, sexuality, ethnicity, a combination of these (i.e., intersectional stigma) or other factors.

The inclusion of a PRISMA flow diagram would enhance the clarity of the methodological approach.

We have included our PRISMA flow diagram.

The article would be strengthened by the inclusion of a conclusion section, a synthesis of key findings, and recommendations for future research in this area.

Pages 17 through 20 conclude the report, and they include i) a synthesis of key findings vis a vis our four research objectives in the 6 paragraphs following the heading Discussion. Suggestions for future research (and for practice) are presented in paragraphs 1 and 2 on page 19.

I would like a bit more rationale for why you chose to focus on Indigenous Peoples in Canada, and not other countries.

Our study emphasizes the cultural nature of health-related stigma, and the Indigenous Peoples in Canada, comprising over 600 distinct communities, encompassing many different languages and ways of life that respond to remarkably different physical environments, presents an excellent opportunity to explore cross-cultural constructions of illness. At the same time, our selection bounded the study in ways that made it feasible and conceptually coherent.

For international readers, you may want to include in the introduction about the inter-generational trauma experienced by this group and how this could impact both health and stigma and health-seeking behaviours.

We have implemented this important revision and described it in our response to a similar request in number point 9 (above). The new information is added to the stipulation in our original submission that:

“ . . . we excluded articles in which the participants’ experiences of stigma stemmed primarily from qualities other than their health condition; most frequently, these included the participants’ occupation, sexual orientation, gender, and cultural or ethnic identity. Stigma that arises at the intersection of these factors is a pressing topic for research; however, it was not the focus of this review.”

For the inclusion criteria, you stated you included primary research articles, but you did not stipulate whether you have any restriction on methods of the primary research articles (i.e. only qualitative articles)? If you did, this should be stated in the inclusion criteria.

We did not have any restrictions on “qualitative” or “quantitative” studies, and we have added a short qualification in the pertinent text in the Methods section. With the revision, it now reads: “We included only primary research (both post-positivist and interpretivist), and excluded articles categorized as essays, commentaries, position statements, letters, or reviews.”

I would like a table title and number for all the tables in the manuscript.

Table 2. Characteristics of included studies (p. 10) lists all of the studies included in the manuscript, along with the relevant data that was extracted from each. Supporting Information S3 Table Articles assessed for eligibility presents the complete list of 997 articles that underwent fulltext screening.

If some articles had a definition of stigma, this should be included in the data extraction. If none of them had a definition of stigma, this should be stated.

Some articles presented a definition of stigma and this was included in the data extraction. S3 Table, which is included for readers as supporting information, includes all of the data that was abstracted from the studies used in the scoping review, including the definitions of stigma.

---

## [Editor Report · Decision Letter 1]

19 Jan 2025

Health-Related Stigma Among Indigenous Peoples in Canada: A Scoping Review

PONE-D-24-51625R1

Dear Dr. Liam Rourke,

We’re pleased to inform you that your manuscript has been judged scientifically suitable for publication and will be formally accepted for publication once it meets all outstanding technical requirements.

Kind regards,

Julia Morgan

Academic Editor

PLOS ONE
---

## [Editor Report · Acceptance letter]

PONE-D-24-51625R1

PLOS ONE

Dear Dr. Rourke,

I'm pleased to inform you that your manuscript has been deemed suitable for publication in PLOS ONE. Congratulations! Your manuscript is now being handed over to our production team.

Kind regards,

on behalf of

Dr. Julia Morgan

Academic Editor

PLOS ONE